# Purity control of simulated moving bed based on advanced fuzzy II controller

**Chao-Fan Xie** [ID]¹ᵒ, **Ting Lin**¹ᵒ, **Hong Zhang**²ᵒ*

1 Department of Big Data and Artificial Intelligence, Fujian Polytechnic Normal University, Fuzhou, China,
2 Key Laboratory of Nondestructive Testing, Fujian Polytechnic Normal University, Fuzhou, China

ᵒ These authors contributed equally to this work.
* zhhgw@hotmail.com

**Data Availability Statement:** All relevant data are within the paper and its Supporting Information files.

## Abstract

Simulated Moving Bed (SMB) is the optimal technology for chromatographic separation, but its process is complex and sensitive to numerous parameters that affect separation performance, making it difficult to control. In recent years, fuzzy controllers have been widely applied in industry due to their simplicity, robustness, and ease of implementation. However, traditional fuzzy controllers used in industry do not consider the error acceleration term. In steady-state conditions, error acceleration is typically slightly less than the target value. Introducing the acceleration term, albeit non-fuzzy, in a proactive fuzzy I-type controller often leads to an increase in steady-state values. The study shows that, compared to the advanced fuzzy I-type controller, the extraction accuracy for material B improved by an average of 0.7%, while the accuracy for material A increased by 0.1%. Compared to traditional fuzzy controllers, the extraction accuracy for material B improved by an average of 0.35%, while the accuracy for material A remained relatively stable. In terms of stability analysis concerning variations in moving bed parameters, the advanced fuzzy II-type controller exhibited greater stability than the I-type, with an average precision stability improvement of 0.6%. Traditional fuzzy controllers demonstrated pathological characteristics during fluctuations in the switching time parameter, whereas the advanced fuzzy II type controller-maintained stability.

## 1 Introduction

The Simulated Moving Bed (SMB) process is a continuous chromatographic separation method simulating countercurrent motion of solid-liquid phases. The traditional SMB setup consists of several packed bed columns arranged in a ring configuration, divided into four zones by two inlet ports (feed and desorb) and two outlet ports (extract and raffinate), as depicted in Fig 1. Ports are shifted in the same direction as the liquid phase flow to simulate countercurrent movement of the solid phase. The feed mixture is continuously introduced into the process through the feed port. Components of the mixture are separated within the columns and collected at the two outlet ports [1–3].

**Funding:** This research was funded by the National Natural Science Foundation of China under Grant 62071123, 61601125), by Natural Science Foundation of Fujian Province of China (number: 2023J011117), by the Fujian Province Education Hall Youth Project (number: JAT220258), by the Fujian Natural Science Foundation Project (number: 2024J01971, 2019J01887).

**Competing interests:** NO authors have competing interests

**Fig 1. Simulated moving bed operation process.**

In terms of controller design, they are specific to particular materials with no universal controller available. Based on the practical moving bed model (TMB), the method utilizes input-output linearization principles [4] and references nonlinear state estimators in the design of SMB control mechanisms. Chao-fan xie et al. utilized an advanced I fuzzy controller to regulate SMB systems, demonstrating its superiority over traditional fuzzy controllers [5]. Santos R.V.A et al. proposed a global parameter estimation method, which estimates the parameters of the SMB system and precisely identifies the model, providing a foundation for the selection of subsequent control strategies [6]. Gillarová et al. developed programs and equipment that can be further applied to the separation of liquid mixtures of various carbohydrates. They identified the optimal operating conditions for the hydrolysis and separation steps of the entire technology. The results provide a foundational dataset for the further development of industrial applications [7]. Based on the theory of triangles, Frandsen et al. proposed a method for the design and validation of isometric gradient chromatography. The results demonstrated the method's effectiveness in enhancing SMB performance and can serve as an initial step in designing and optimizing operational flow rates and modifier concentrations [8]. Woo-Sung Lee et al. proposed a machine learning algorithm for improving the purity of n-alkane extraction. The study highlights the significance of zone flushing in impurity removal, an aspect that cannot be fully estimated from dynamic models [9]. Reinaldo et al. proposed a variant of the SMB system characterized by adjustable column segment lengths and feed concentrations. These features offer greater flexibility compared to traditional operations, leading to fundamental improvements in the separation and purification of mixtures [10]. Marrocos et al. proposed a deep artificial intelligence architecture for online soft sensing, featuring a nonlinear output error framework and a nonlinear autoregressive predictor with external inputs. This structure provides insights into predicting the main characteristics of simulated moving bed

chromatography equipment [11]. Hoon et al. proposed a deep Q-network (DQN) based on a model-free reinforcement learning method to train a control strategy for the SMB process in a data-driven manner, approaching optimality [12]. Further studies on other SMB controllers can be found in references [13–20].

Overall, these studies aim to separate specific industrial equipment and materials. The cost of conducting experiments based on physical machinery is relatively high, and the developed controllers lack generalizability. As environmental parameters change, the results can easily lead to separation failures. To enhance the performance of SMB, precise physical parameters of the moving bed must be obtained through laborious experiments. However, the physical parameters derived from experiments may not fully represent the actual operation of the SMB system. Due to factors such as connections between pipes or the long-term use of solid phases, parameters of the entire SMB may vary, leading to discrepancies between physical performance and experimental parameters. These issues can result in insufficient operating conditions for the SMB, causing unsatisfactory chromatographic separation. Generally, the SMB system exhibits variability during operation. Optimal control of the SMB can be achieved by online monitoring the real-time concentration of substances and immediately adjusting the SMB's operating conditions. The foundation of this study is the development of a general fuzzy controller based on the discrete dynamic system of SMB. Preliminary research involved traditional fuzzy controllers and advanced fuzzy I-type controllers, but their accuracy and stability were insufficient. Building upon this, the advanced fuzzy II type controller proposed in this study demonstrates improved performance compared to the previous two types of fuzzy controllers.

## 2 SMB mathematical model

For a four-zone closed-loop simulated moving bed (SMB) unit, the researchers developed a generalized mathematical model following the principles from dynamical systems theory [21]. The model assumes a classical setup for materials and parameters, where solvents, extractants, feed, and pumps at the inlets and outlets of each zone (I, II, III, and IV) separate the respective segments. The fixed-bed adsorption model serves as the foundation for the SMB model. The liquid phase mass balance for the $i$-th component in the $j$-th column is given by the following equation:

$$\frac{\partial C_{ij}}{\partial t} = D_i \frac{\partial^2 C_{ij}}{\partial x^2} - v_j^* \frac{\partial C_{ij}}{\partial x} - \frac{1-\varepsilon}{\varepsilon} k_i (q_{ij}^* - q_{ij}) \tag{1}$$

$$\frac{\partial q_{ij}}{\partial t} = k_i (q_{ij}^* - q_{ij}) \tag{2}$$

With initial and boundary condition is [21]:

$$C_{ij}(x, \, 0) = C_{0ij} \tag{3}$$

$$\frac{\partial C_{ij}(x,t)}{\partial x}\big|_{x=L_{end}} = 0 \tag{4}$$

$$D_i \frac{\partial C_{ij}(x,t)}{\partial x}\big|_{x=L_0} = v_j^* [C_{ij}(L_0, t) - \bar{C}_{ij}^{\,sect}(t)] \tag{5}$$

**Table 1. Parameters of SMB system.**

| Parameter | Nomenclature | Parameter | Nomenclature |
|---|---|---|---|
| $x(cm)$ | Axial distance | $Q(cm^3min^{-1})$ | Volume flow rate |
| $k(gL^{-1})$ | Comprehensive mass transfer constant | $t(second)$ | Time |
| $v^*(cm\,min^{-1})$ | Effect velocity of body | $D(cm^2min^{-1})$ | Effective dispersion coefficient |
| $us(cm\,min^{-1})$ | Solid flow rate | $\varepsilon$ | Bulk void fraction |
| $C(gL^{-1})$ | Mobile phase concentration | $i$ | Material index: A or B |
| $q(gL^{-1})$ | Solid phase concentration | $j$ | Column number: 1, 2, 3, 4, 5, 6, 7, 8 |
| $q^*(gL^{-1})$ | Solid phase concentration at equilibrium between solid phase and mobile phase | $C_{0ij}$ | Initial concentration distribution |
| $L_{end}$ | The end position of column | $L_0$ | The initial position of column |
| $C_{fi}$ | The concentration of the material import | $\bar{C}_{ij}^{\ sect}$ | Described in Eqs (12–14) |
| $C_{E,B}$ | Concentration of material B | $C_{R,A}$ | Concentration of material A |
| $\bar{C}_{E,B}$ | Purity of material B | $\bar{C}_{R,A}$ | Purity of material A |

The parameters meaning is shown in Table 1. $\bar{C}_{ij}^{\ sect}(t)$ term is related to the located region; it be divided into three cases shown in the following equations [21]:

$$\bar{C}_{ij}^{\ I}(t) = \frac{Q_{IV}\,C_{ij-1}(L_{end},\,t)}{Q_I},\,Section\,I,\,1^{st}column \tag{6}$$

$$\bar{C}_{ij}^{\ III}(t) = \frac{Q_{II}\,C_{ij-1}(L_{end},\,t) + Q_f C_{fi}}{Q_{III}}\,Section\,III,\,1^{st}column \tag{7}$$

$$\bar{C}_{ij}^{\ sect}(t) = C_{ij-1}(L_{end},\,t), other \tag{8}$$

Eq (2) is substituted into (1), it can get:

$$\frac{\partial C_{ij}}{\partial t} = D_i \frac{\partial^2 C_{ij}}{\partial x^2} - v_j^* \frac{\partial C_{ij}}{\partial x} - \frac{1-\varepsilon}{\varepsilon}\frac{\partial q_{ij}}{\partial t} \tag{9}$$

The adsorption equilibrium for both two enantiomers is represented by linear isotherms [22]:

$$q_{ij} = H_i C_{ij} \tag{10}$$

Purity equations are:

$$\bar{C}_{E,B} = \frac{C_{E,B}}{C_{E,A} + C_{E,B}} \tag{11}$$

$$\bar{C}_{R,A} = \frac{C_{R,A}}{C_{R,A} + C_{R,B}} \tag{12}$$

## 3 Simulation

The simulated physical system comprises 8 packed columns arranged in a 2-2-2-2 configuration. Parameters for the columns, equilibrium, and mass transfer can be found in Table 2, with values referenced from relevant literature [21]. The adsorption equilibrium of the two materials is represented by linear isotherms. The 16 partial differential equations (PDEs) were

**Table 2. The initial parameters for SMB.**

| Parameter | Value | Parameter | Value |
|---|---|---|---|
| $L(cm)$ | 25 | $C_{f,i}(gL^{-1})$ | 5 |
| $d(cm)$ | 0.46 | $\theta(min)$ | 3 |
| $H_A$ | 0.001 | $Q_I(cm^3min^{-1})$ | 6.75 |
| $H_B$ | 0.45 | $Q_{II}(cm^3min^{-1})$ | 6.6 |
| $D_A(cm^2min^{-1})$ | 0.2 | $Q_{III}(cm^3min^{-1})$ | 7 |
| $D_B(cm^2min^{-1})$ | 1.265 | $Q_{IV}(cm^3min^{-1})$ | 2 |
| $\varepsilon$ | 0.8 | spatial number | 50 |

discretized using the Crank-Nicolson finite difference method on a mesh with 600 elements. The time step was 0.1 seconds, each column was divided into 50 segments, and the spatial step size was 0.5 cm, resulting in an implicit linear system with 400 unknowns. The linear system was solved using the function library in MATLAB 2019, leveraging sparse matrix algebra algorithms due to the triangularly sparse nature of the matrix.

## 3.1 Advance fuzzy II controller design

Adding the input term of error acceleration to the traditional fuzzy controller and fuzzifying it into linguistic variables is a feature of the advanced fuzzy II-type controller. Thus, in the advanced fuzzy II-type controller, the input variables include error, first-order error difference, and second-order error, formulated as follows:

$$e_1 = desired\ B - C_{E,B} \tag{13}$$

$$e_2 = desired\ A - C_{R,A} \tag{14}$$

$$e_3 = e_1 + e_2 \tag{15}$$

$$\Delta e_1 = e_1(k) - e_1(k-1) \tag{16}$$

$$\Delta e_2 = e_2(k) - e_2(k-1) \tag{17}$$

$$\Delta e_3 = \Delta e_1 + \Delta e_2 \tag{18}$$

$$\Delta^2 e_1 = \Delta e_1(k) - \Delta e_1(k-1) \tag{19}$$

$$\Delta^2 e_2 = \Delta e_2(k) - \Delta e_2(k-1) \tag{20}$$

$$\Delta^2 e_3 = \Delta^2 e_1 + \Delta^2 e_2 \tag{21}$$

Where $e_i$, $\Delta e_i$, $\Delta^2 e_i$ are the input varialbles of the zone $i$ ($i$ = 1,2,3) controller. The control structure of advanced fuzzy II controller in SMB system is shown in Fig 2.

The fuzzy system defined five linguistic variable values for the input variables, including error, first-order error, and second-order error: *NB* (Negative Big), *NS* (Negative Small), *ZE* (Zero), *PS* (Positive Small), and *PB* (Positive Big). The membership functions are illustrated in Fig 3. Subsequently, the control parameter flow was defined $\Delta Q_I, \Delta Q_{II}, \Delta Q_{III}$, taking a unipolar fuzzy value, the values are shown in Table 3.

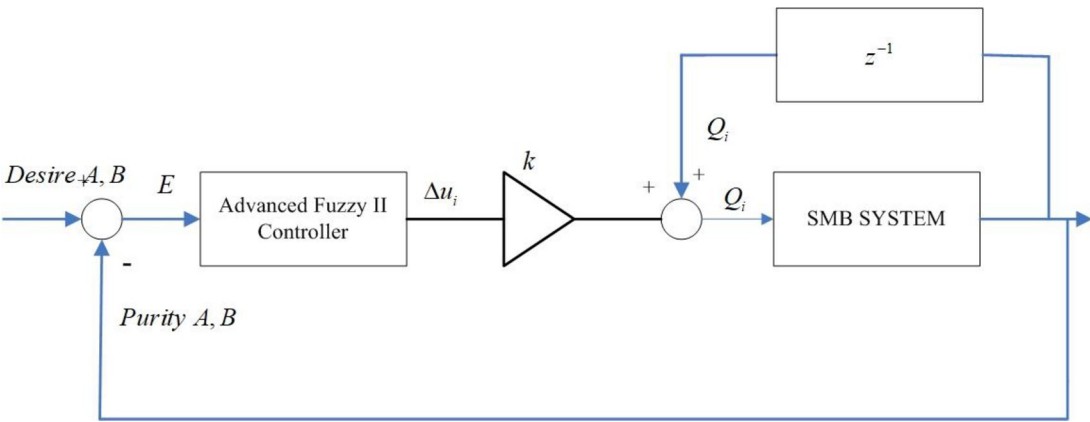

**Fig 2. Advanced fuzzy II controller for SMB.**

Based on the principles of kinematics, we formed the following rule base Tables 4–6, much like how one can balance a rod on their palm without it falling without needing to know Newton's equations of motion. The simple rules followed are the foundation of this fuzzy rule base, allowing us to bypass the complex inference of control equations.

The fuzzy inference engine employs the product form, where the strength of the If part is multiplied by the output value of the Then part of the fuzzy rule [23]. The final step is defuzzification. Here, we choose the center of weighted average method for decomposition. From the fuzzy rule base and the membership functions of the input variables, it can be observed that each input activates four rules. The formulas are shown in (22), (23), and (24).

$$
\begin{aligned}
F_{if-then} = {} & T(u_1, u_3, u_5) \times y_1 + T(u_1, u_3, u_6) \times y_2 + T(u_2, u_3, u_5) \times y_3 + T(u_2, u_3, u_6) \times y_4 \\
& + T(u_1, u_4, u_5) \times y_5 + T(u_1, u_4, u_6) \times y_6 + T(u_2, u_4, u_5) \times y_7 + T(u_2, u_4, u_6) \times y_8
\end{aligned} \tag{22}
$$

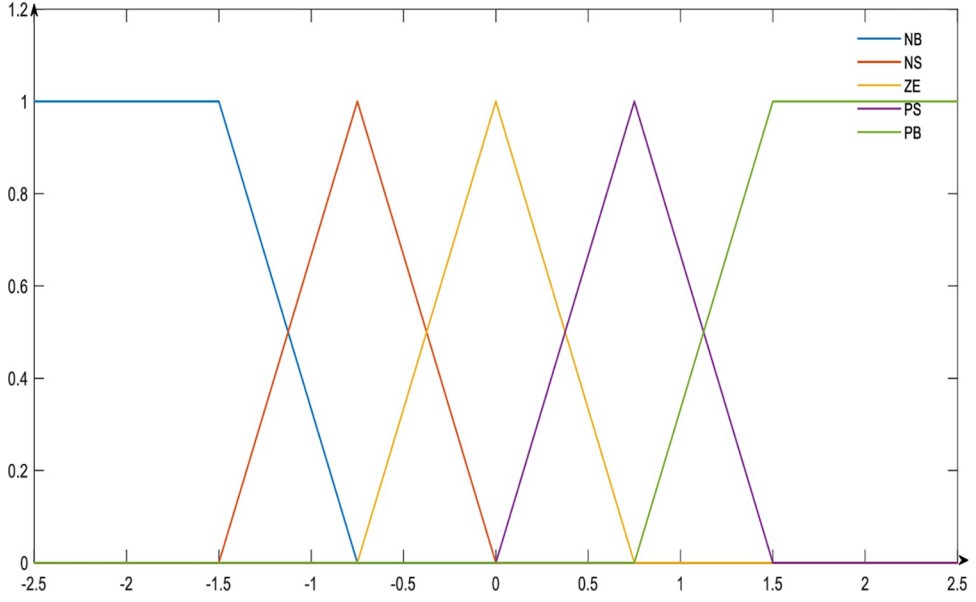

**Fig 3. The memberships function for $e_i$, $\Delta e_i$ and $\Delta^2 e_i$.**

**Table 3. Unipolar fuzzy values for ΔQ$_{I}$, ΔQ$_{II}$, ΔQ$_{III}$.**

| Region | A | B | C | D | E | F | G | H | I |
|--------|-----|-----|-----|-----|-----|-----|-----|-----|-----|
| ΔQ$_{I}$ | -0.18 | -0.12 | -0.08 | -0.05 | 0 | 0.05 | 0.08 | 0.12 | 0.18 |
| ΔQ$_{II}$ | -0.0007 | -0.0006 | -0.0005 | -0.0004 | 0 | 0.0004 | 0.0005 | 0.0006 | 0.0007 |
| ΔQ$_{III}$ | -0.12 | -0.1 | -0.07 | -0.04 | 0 | 0.04 | 0.07 | 0.1 | 0.12 |

$$W = T(u_1, u_3, u_5) + T(u_1, u_3, u_6) + T(u_2, u_3, u_5) + T(u_2, u_3, u_6)$$
$$+ T(u_1, u_4, u_5) + T(u_1, u_4, u_6) + T(u_2, u_4, u_5) + T(u_2, u_4, u_6) \tag{23}$$

$$U = \frac{F_{if-then}}{W} \tag{24}$$

Variables $u_1, u_2$ represent fuzzy error acceleration, corresponding to second-order differences, while variables $u_3, u_4$ denote fuzzy error values, and variables $u_5, u_6$ signify fuzzy error velocity, corresponding to first-order differences. Variable $y_1 \cdots y_8$ corresponds to monopolar values of eight activated rules, representing the fuzzy monopolar values in the Then part. Table 5 includes only four rules; the other four rules are considered as exceptions with a value of 0. The symbol $T$ denotes the product $T$-norm, and the final output value $U$ is determined by the defuzzifier.

## 3.2 Purity control experiment

In this section, we use an advanced II fuzzy controller to regulate the purity of the SMB system. Figs 4 and 5 illustrate the control results. Regarding the precision of the controller in achieving control targets, we designed two sets of experiments with purity targets of 94% and 96% for material B, and 96% and 94% for material A. The relevant control results are displayed in Table 7, and the control effect diagrams are shown in Figs 4 and 5.

From the outlet of the extracted liquid in Fig 4 and Table 7, in the first set of experiments, the purity targets for controlling material B and material A were set at 94% and 96%, respectively. For material B, the final control results showed that the advanced fuzzy type II controller achieved 94%, the type I controller achieved 94.78%, and the traditional controller achieved 93.13%. The precision of the type II controller was 100%, the type I controller was 99.2%, and the traditional controller was 99.8%. For material A, the type II controller achieved 95.94%, the type I controller achieved 96.14%, and the traditional controller achieved 96.02%. The precision of the type II controller was 99.9%, the type I controller was 99.8%, and the traditional controller was 99.9%.

It is evident that the advanced fuzzy II controller provides the best accuracy and does not exhibit the minor oscillations seen with traditional fuzzy control. The advanced fuzzy

**Table 4. $\Delta^2 e(k) > 0$ assignment table.**

| Δ²e | PS | PB | PS | PB | PS | PB | PS | PB | PS | PB |
|-----|----|----|----|----|----|----|----|----|----|----|
| e | NB | | NS | | ZE | | PS | | PB | |
| Δe | | | | | | | | | | |
| NB | A | A | A | A | A | A | C | B | D | C |
| NS | A | A | B | A | C | B | E | D | F | E |
| ZE | A | A | C | C | E | E | G | G | I | I |
| PS | E | F | F | G | H | I | I | I | I | I |
| PB | F | G | H | I | I | I | I | I | I | I |

**Table 5. $\Delta^2(k) = 0$ assignment table.**

| $\Delta e$ \\ $e$ | NB | NS | ZE | PS | PB |
|---|---|---|---|---|---|
| NB | A | A | A | C | E |
| NS | A | B | C | E | G |
| ZE | A | C | E | G | I |
| PS | C | E | G | H | I |
| PB | E | G | I | I | I |

controller also shows minor fluctuations but has a slower convergence rate compared to type II, which is the trade-off for increased accuracy. At the raffinate liquid outlet, both advanced fuzzy II controller and traditional fuzzy controller demonstrate better control precision than advanced fuzzy I controller; however, the fluctuations of type II are somewhat larger compared to the traditional controller.

In the experiments shown in Fig 5 and Table 7, in the second set of experiments, the purity targets for controlling material B and material A were set at 96% and 94%, respectively. For material B, the final control results showed that the advanced fuzzy type II controller achieved 96%, the type I controller achieved 96.5%, and the traditional controller achieved 95.43%. The precision of the type II controller was 100%, the type I controller was 99.4%, and the traditional controller was 99.5%. For material A, the type II controller achieved 94.04%, the type I controller achieved 94.11%, and the traditional controller achieved 94.01%. The precision of the type II controller was 99.9%, the type I controller was 99.8%, and the traditional controller was 99.9%.

At the extract liquid outlet, the advanced fuzzy II controller still provides the best accuracy. In addition to its relatively slow convergence speed, it also does not exhibit oscillations or fluctuations. However, at the raffinate liquid outlet, despite its high accuracy, the type II controller shows greater fluctuations compared to both the advanced fuzzy I controller and the traditional fuzzy controller.

Taking the control objective illustrated in Fig 4 as an example, the following Figs 6 and 7 shows the progression of force exerted during the control process. It can be observed that the control equipment needs to frequently switch control directions during the process. In practical applications, the requirements for the control equipment are relatively high, as the controller must achieve smooth transitions during these direction changes.

## 3.3 Stability experiment

In this section, we will examine the effects of changes in adsorbent parameters, feed concentration, and switching times on the controller's performance. The control results are illustrated in Figs 8–10.

**Table 6. $\Delta^2 e(k) < 0$ assignment table.**

| $\Delta e$ \\ $\Delta^2 e$ | NS | NB | NS | NB | NS | NB | NS | NB | NS | NB |
|---|---|---|---|---|---|---|---|---|---|---|
| $e$ | NB | | NS | | ZE | | PS | | PB | |
| NB | A | A | A | A | A | A | B | A | D | C |
| NS | A | A | B | A | B | A | D | C | E | D |
| ZE | A | A | C | C | E | E | G | G | I | I |
| PS | D | E | E | F | G | H | H | I | I | I |
| PB | E | F | G | H | I | I | I | I | I | I |

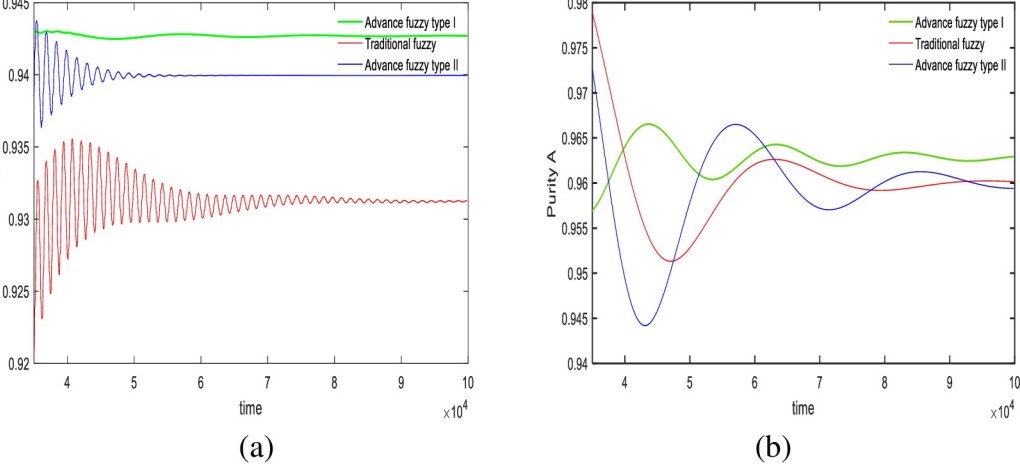

**Fig 4. Desired *B* = 94%, desired *A* = 96%, Switch time = 180s.**

Fig 8 shows the impact of variations in adsorbent parameters on the controller's performance, with two sets of experiments targeting purities of 92% and 96%. The experimental results indicate that at the extracted liquid outlet, the advanced fuzzy II controller is the most stable, exhibiting the least amount of oscillation and fluctuation. At the raffinate liquid outlet, In this outlet, the two sets of experiments controlled purities of 93% and 97%, the control precision of the three controllers is similar. However, due to the inclusion of acceleration terms, both advanced fuzzy I and II controllers exhibit greater fluctuations compared to the traditional fuzzy controller.

Fig 9 illustrates the impact of feed concentration on control performance. At the extracted liquid outlet, it is observed that the traditional fuzzy controller exhibits significant oscillations, while both advanced fuzzy controllers are relatively stable, with type II providing better control precision. At the raffinate liquid outlet, for a purity of 93%, type II controller performs better; however, for a purity of 97%, the results from the type II fuzzy controller are slightly overestimated. Overall, advanced fuzzy II controller demonstrates greater effectiveness.

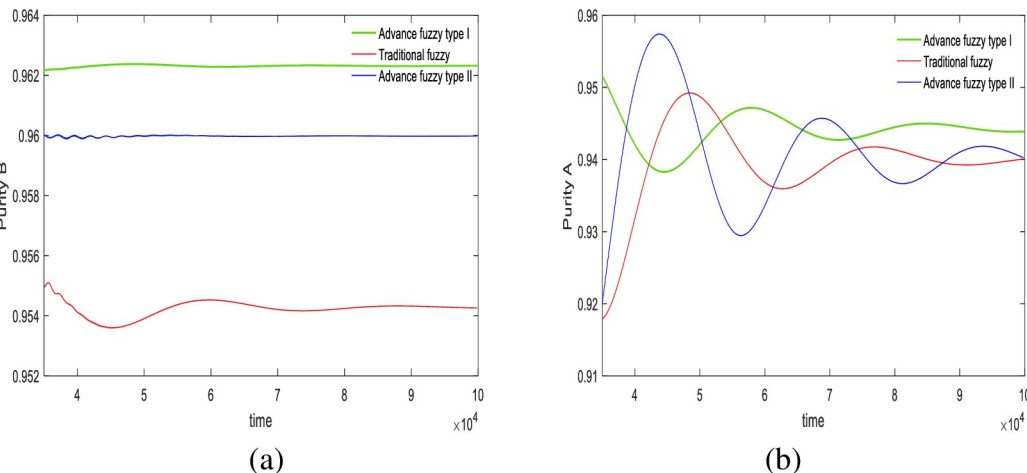

**Fig 5. Desired *B* = 96%, desired *A* = 94%, Switch time = 180s.**

**Table 7. First and second group experiments.**

| Group \ name | Target | Advance fuzzy type II | Advance fuzzy type I | Traditional fuzzy |
|---|---|---|---|---|
| Group1 Material A | 96% | 95.94% | 96.14% | 96.02% |
| Group1 Material B | 94% | 94% | 94.78% | 93.13% |
| Group2 Material A | 94% | 94.04% | 94.11% | 94.01% |
| Group2 Material B | 96% | 96% | 96.5% | 95.43% |

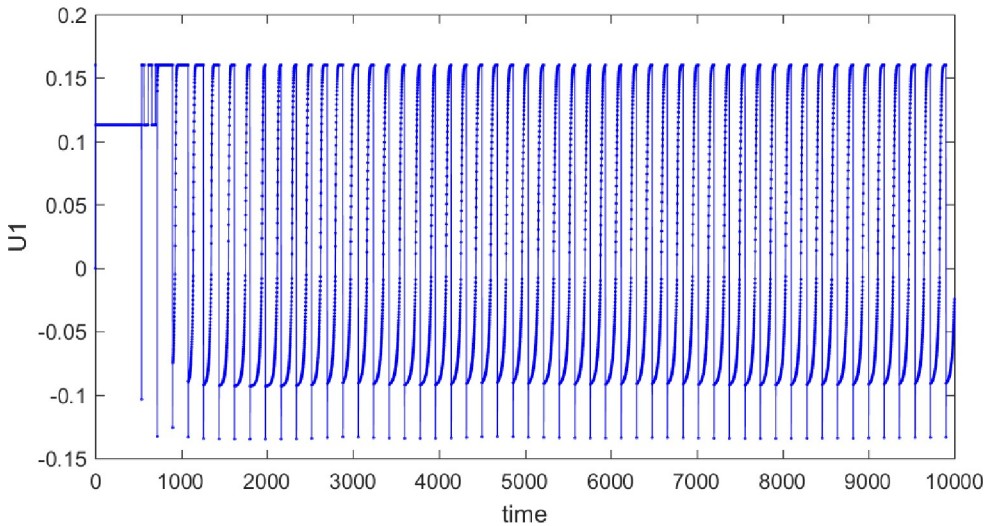

**Fig 6. Diagram of force application in the purity control process of material B.**

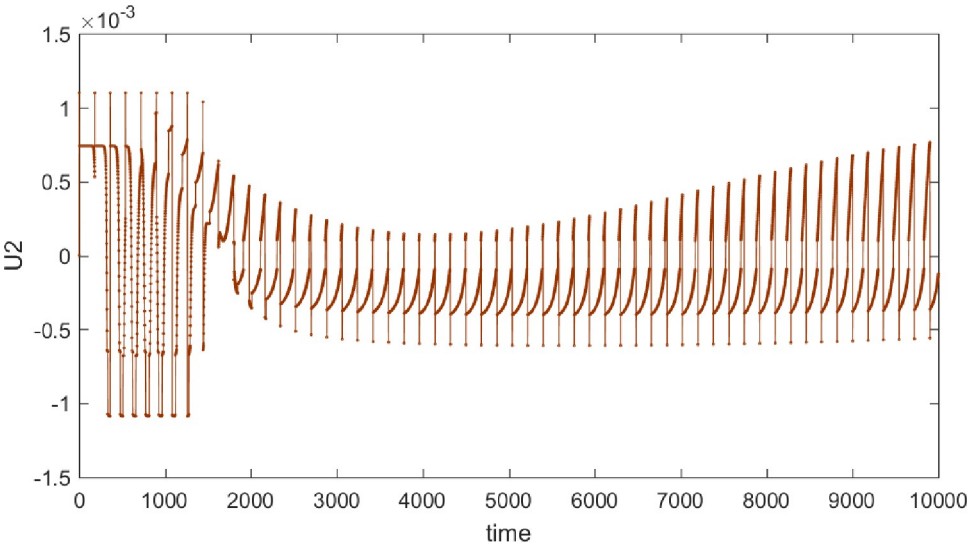

**Fig 7. Diagram of force application in the purity control process of material A.**

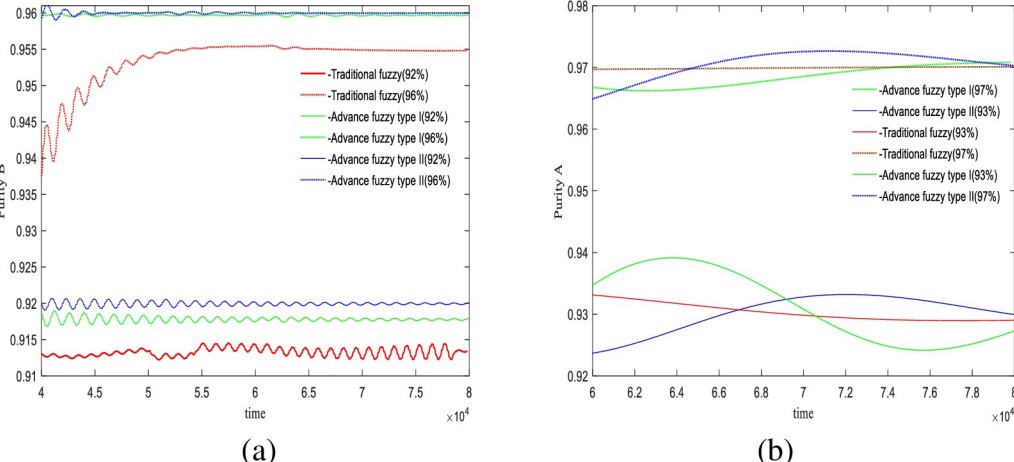

**Fig 8. Under the change of adsorbent parameters $H_A$ = 0.01→0.03.**

The final set of experiments observed the impact of switching time on control results shown in Fig 10. At both the extracted liquid outlet and the raffinate liquid outlet, the traditional fuzzy controller exhibited pathological characteristics. Among the advanced fuzzy controllers, type II demonstrated higher precision than type I; however, it also exhibited relatively higher fluctuation and slower convergence compared to type I.

## 4 Conclusion

This study primarily investigates the use of advanced fuzzy II controller for regulating the purity of simulated moving bed separations. The research demonstrates that compared to the traditional fuzzy controller and advanced fuzzy I controller, advanced fuzzy II controller achieves higher control precision, albeit at the cost of slower convergence. The advanced fuzzy II controller also maintains stability and high precision when subjected to variations in adsorbent parameters, feed concentration, and switching time. Particularly under switching time variations, advanced fuzzy II controller remains stable, though it exhibits slightly greater fluctuations compared to type I. Compared to the traditional fuzzy controller, the extraction

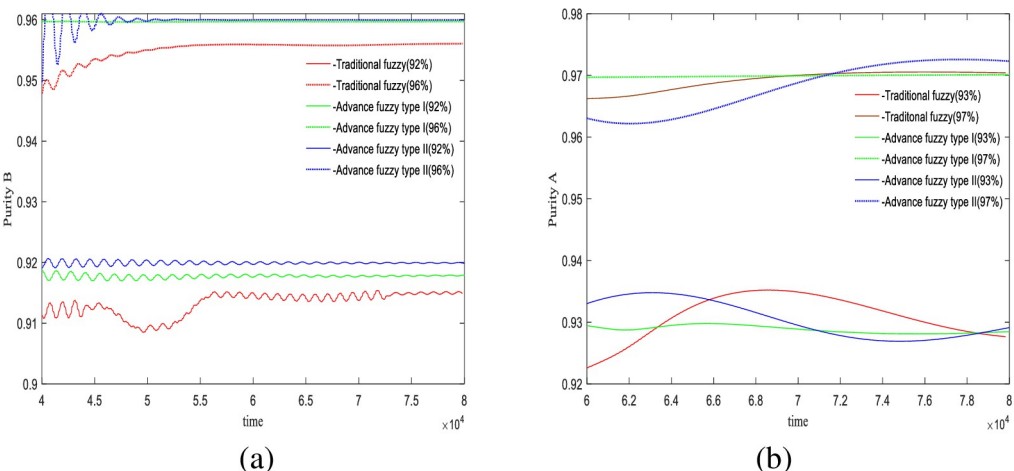

**Fig 9. Under the change of feed port concentration $C_f$ = 4.5→5.2.**

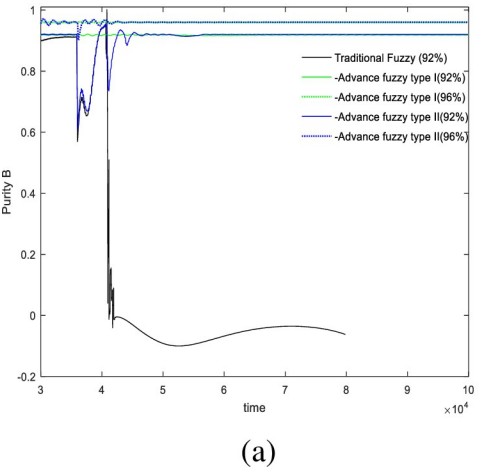
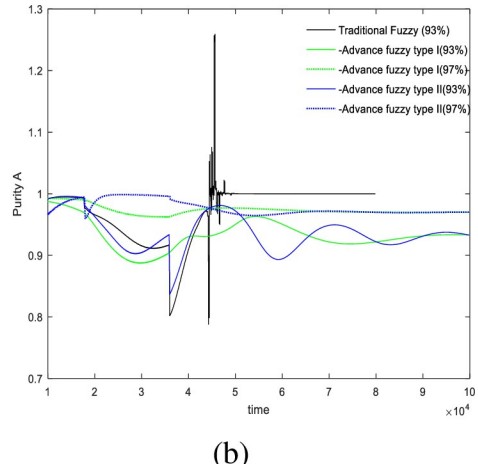

(a)                                    (b)

**Fig 10. Under the change of switch time $\theta = 178s \rightarrow 182s$.**

accuracy of Material B improved by an average of 0.35%, while the accuracy of Material A remained relatively stable. In the stability analysis of moving bed parameter variations, the advanced Type II fuzzy controller exhibited greater stability than the Type I controller, with an average precision stability improvement of 0.6%. Overall, advanced fuzzy II controller proves to be superior to both the traditional fuzzy controller and advanced fuzzy I controller.

Theoretically, the advanced fuzzy II controller offers a novel approach for controlling highly sensitive nonlinear systems, potentially improving control precision by appropriately increasing the fuzzy control quantities in the dynamics. In practical applications, higher precision control can reduce the frequency of experiments and adjustments, thereby lowering production costs. Additionally, it ensures product consistency and high quality. By precisely controlling various parameters in the separation process, the controller can better manage product quality metrics and meet market demands.

## 5 Future directions

In the realm of scientific research, the advanced fuzzy type II controller generally demonstrates better accuracy and stability. However, it is evident that all three controllers perform suboptimal with respect to the parameter adjustment of switching time. Given the strong learning capability of neural networks, future directions could involve combining neural networks with advanced fuzzy controllers to enhance the adaptability of the controllers. In terms of application, it would be beneficial to experiment with model predictive control on discrete dynamic systems in simulated moving beds, and subsequently adjust parameters to apply the advanced fuzzy type II controller in real production settings to observe actual control efficiency. The significance of achieving intelligent control for simulating moving beds is substantial.

## Supporting information

**S1 Data.**
(RAR)

## Author Contributions

**Conceptualization:** Hong Zhang.

**Data curation:** Chao-Fan Xie, Ting Lin.

**Formal analysis:** Chao-Fan Xie, Ting Lin.

**Investigation:** Hong Zhang.

**Project administration:** Hong Zhang.

**Software:** Chao-Fan Xie, Ting Lin.

**Supervision:** Hong Zhang.

**Validation:** Ting Lin.

**Writing – original draft:** Chao-Fan Xie.

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
