## [Decision Letter · Decision Letter 0]

11 Sep 2024

PONE-D-24-28958Purity Control of Simulated Moving Bed Based on Advanced Fuzzy II ControllerPLOS ONE

Dear Dr. xie,

Thank you for submitting your manuscript to PLOS ONE. After careful consideration, we feel that it has merit but does not fully meet PLOS ONE’s publication criteria as it currently stands. Therefore, we invite you to submit a revised version of the manuscript that addresses the points raised during the review process.

We look forward to receiving your revised manuscript.

Kind regards,

Dr Satya Veerendra Arigela, Ph.D.

Academic Editor

PLOS ONE

**Journal Requirements:**

- Shortcut design method for multicomponent gradient simulated moving beds (DOI: 10.1002/aic.18304)

(among others)

In your revision ensure you cite all your sources (including your own works), and quote or rephrase any duplicated text outside the methods section. Further consideration is dependent on these concerns being addressed.

This research was partially supported by the National Natural Science Foundation of China under Grant 62071123，61601125), by Natural Science Foundation of Fujian Province of China (number: 2023J011117), by the Fujian Province Education Hall Youth Project (number: JAT220258), by the Fujian Natural Science Foundation Project (number: 2019J01887).

5. In the online submission form you indicate that your data is not available for proprietary reasons and have provided a contact point for accessing this data. Please note that your current contact point is a co-author on this manuscript. According to our Data Policy, the contact point must not be an author on the manuscript and must be an institutional contact, ideally not an individual. Please revise your data statement to a non-author institutional point of contact, such as a data access or ethics committee, and send this to us via return email. Please also include contact information for the third party organization, and please include the full citation of where the data can be found.

7. Please amend either the abstract on the online submission form (via Edit Submission) or the abstract in the manuscript so that they are identical.

**Additional Editor Comments:**

The contribution of this manuscript is not clear. Therefore, my recommendation is to accept (Major Revision) this manuscript that has Ref. No.: PONE-D-24-28958 after several adjustments must be made before publication.

My specific comments are:

1- In the abstract, the result of this work must be described briefly with data in order to show the effectiveness of the proposed work.

2- The author did not describe the drawbacks of each conventional technique in the introduction paragraph.

3- Please include the references for all equations.

4- Please put a figure that describes the block diagram of the closed loop system and put in the figure what is the output variable and the desired output?

5- Please add figures that demonstrate the control action for all cases.

6- In conclusions, please add an enhancement percentage (%) that demonstrates the proposed algorithm efficiency for your method when used with another method.

Reviewers' comments:

Reviewer's Responses to Questions

**Comments to the Author**

1. Is the manuscript technically sound, and do the data support the conclusions?

Reviewer #1: Yes

Reviewer #2: Partly

2. Has the statistical analysis been performed appropriately and rigorously? 

Reviewer #1: Yes

Reviewer #2: Yes

3. Have the authors made all data underlying the findings in their manuscript fully available?

Reviewer #1: Yes

Reviewer #2: No

4. Is the manuscript presented in an intelligible fashion and written in standard English?

Reviewer #1: Yes

Reviewer #2: Yes

5. Review Comments to the Author

**Reviewer #1: **Firstly, I would like to congratulate the authors for the excellent work so far, below are some suggestions to qualify the research, namely:

- Please describe in more detail the Summary of this research, thus highlighting its importance to the scientific and academic community;

- Please explain in more detail the figures presented in the article, it highlights their importance for the present work;

- I suggest inserting one more topic, right after the Conclusion, addressing what is expected from this work in the future. This aspect is essential as it demonstrates the importance of the present study for the future;

https://ieeexplore.ieee.org/abstract/document/9758783

https://link.springer.com/chapter/10.1007/978-3-031-42685-8_4

https://link.springer.com/chapter/10.1007/978-3-031-05767-0_24

https://ieeexplore.ieee.org/abstract/document/9239074

https://link.springer.com/chapter/10.1007/978-3-031-04435-9_17

Grateful.

**Reviewer #2: **The contribution of this manuscript is not clear. Therefore, my recommendation is to accept (Major Revision) this manuscript that has Ref. No.: PONE-D-24-28958 after several adjustments must be made before publication.

My specific comments are:

1- In the abstract, the result of this work must be described briefly with data in order to show the effectiveness of the proposed work.

2- The author did not describe the drawbacks of each conventional technique in the introduction paragraph.

3- Please include the references for all equations.

4- Please put a figure that describes the block diagram of the closed loop system and put in the figure what is the output variable and the desired output?

5- Please add figures that demonstrate the control action for all cases.

6- In conclusions, please add an enhancement percentage (%) that demonstrates the proposed algorithm efficiency for your method when used with another method.

6. PLOS authors have the option to publish the peer review history of their article (what does this mean?). If published, this will include your full peer review and any attached files.

Reviewer #1: **Yes: **Gabriel Gomes de Oliveira

Reviewer #2: No

---

## [Author Response · Author response to Decision Letter 0]

20 Sep 2024

Explanations to editor

Thank you very much for your comments and suggestions. We have revised the manuscript according to the comments and now explain to the editor as follows:

 Question 1: When submitting your revision, we need you to address these additional requirements.Please ensure that your manuscript meets PLOS ONE's style requirements, including those for file naming. 

 Replies 1: Thank you very much for your valuable suggestion. We have made the necessary formatting changes to the title, authors, figures, and tables in accordance with the journal’s requirements.

 Question 2: We noticed you have some minor occurrence of overlapping text with the following previous publication(s), which needs to be addressed:- Shortcut design method for multicomponent gradient simulated moving beds (DOI: 10.1002/aic.18304)

 Replies 2: Thank you very much for your important suggestion. We have modified the overlapping text.

Question 3: Please note that PLOS ONE has specific guidelines on code sharing for submissions in which author-generated code underpins the findings in the manuscript. In these cases, all author-generated code must be made available without restrictions upon publication of the work. 

 Replies 3: I agree to make my code publicly available if the article is accepted.

Question 4: We note that you have provided funding information that is not currently declared in your Funding Statement. However, funding information should not appear in the Acknowledgments section or other areas of your manuscript. We will only publish funding information present in the Funding Statement section of the online submission form.Please remove any funding-related text from the manuscript and let us know how you would like to update your Funding Statement. Currently, your Funding Statement reads as follows:The author(s) received no specific funding for this work.

 Replies 4: Thank you very much. We have remove any funding-related text from the manuscript and modified as: The author(s) received no specific funding for this work.

 Amended statements have added within our cover letter; Could you please make the changes of online form. 

Question 5: In the online submission form you indicate that your data is not available for proprietary reasons and have provided a contact point for accessing this data. Please note that your current contact point is a co-author on this manuscript. According to our Data Policy, the contact point must not be an author on the manuscript and must be an institutional contact, ideally not an individual. Please revise your data statement to a non-author institutional point of contact, such as a data access or ethics committee, and send this to us via return email. Please also include contact information for the third party organization, and please include the full citation of where the data can be found. 

 Replies 5: Regarding the previous statement, our team members did not reach a consensus at that time. However, we now unanimously agree to make the data publicly available.

Question 6: When completing the data availability statement of the submission form, you indicated that you will make your data available on acceptance. We strongly recommend all authors decide on a data sharing plan before acceptance, as the process can be lengthy and hold up publication timelines. Please note that, though access restrictions are acceptable now, your entire data will need to be made freely accessible if your manuscript is accepted for publication. This policy applies to all data except where public deposition would breach compliance with the protocol approved by your research ethics board. If you are unable to adhere to our open data policy, please kindly revise your statement to explain your reasoning and we will seek the editor's input on an exemption. Please be assured that, once you have provided your new statement, the assessment of your exemption will not hold up the peer review process. 

 Replies 6: Our team members agree to make the data available in advance.

Question 7: Please amend either the abstract on the online submission form (via Edit Submission) or the abstract in the manuscript so that they are identical. 

 Replies 7: Thank you very much for your reminder. We will ensure that the online abstract is revised accordingly at the time of submission.

Additional Editor Comments:

Question 1: In the abstract, the result of this work must be described briefly with data in order to show the effectiveness of the proposed work. 

 Replies 1: Thank you very much for your valuable feedback. We have added a brief description of the experimental results using data in the abstract, highlighted in red. The revised sections are as follows:

 The study shows that, compared to the advanced fuzzy I-type controller, the extraction accuracy for material B improved by an average of 0.7%, while the accuracy for material A increased by 0.1%. Compared to traditional fuzzy controllers, the extraction accuracy for material B improved by an average of 0.35%, while the accuracy for material A remained relatively stable. In terms of stability analysis concerning variations in moving bed parameters, the advanced fuzzy II-type controller exhibited greater stability than the I-type, with an average precision stability improvement of 0.6%.

Question 2: The author did not describe the drawbacks of each conventional technique in the introduction paragraph. 

 Replies 2: Thank you again for your professional suggestion. We have add the describe the drawbacks of each conventional technique in the introduction paragraph. The added parts are in page 3 which were marked in red. The changes are as follows:

Overall, these studies aim to separate specific industrial equipment and materials. The cost of conducting experiments based on physical machinery is relatively high, and the developed controllers lack generalizability. As environmental parameters change, the results can easily lead to separation failures. To enhance the performance of SMB, precise physical parameters of the moving bed must be obtained through laborious experiments. However, the physical parameters derived from experiments may not fully represent the actual operation of the SMB system. Due to factors such as connections between pipes or the long-term use of solid phases, parameters of the entire SMB may vary, leading to discrepancies between physical performance and experimental parameters. These issues can result in insufficient operating conditions for the SMB, causing unsatisfactory chromatographic separation. Generally, the SMB system exhibits variability during operation. Optimal control of the SMB can be achieved by online monitoring the real-time concentration of substances and immediately adjusting the SMB’s operating conditions. The foundation of this study is the development of a general fuzzy controller based on the discrete dynamic system of SMB. Preliminary research involved traditional fuzzy controllers and advanced fuzzy I-type controllers, but their accuracy and stability were insufficient. Building upon this, the advanced fuzzy II type controller proposed in this study demonstrates improved performance compared to the previous two types of fuzzy controllers.:

Question 3: Please include the references for all equations. 

Replies 3: Thank you again for your valuable suggestion. We have add the references for some equations which were obtained through research conducted by others. The added parts are in page 2 and 3, which were marked in red.

Question 3: Please include the references for all equations.

Replies 3: Thank you again for your valuable suggestion. We have add the references for some equations which were obtained through research conducted by others. The added parts are in page 2 and 3, which were marked in red. 

Question 4: Please put a figure that describes the block diagram of the closed loop system and put in the figure what is the output variable and the desired output?

Replies 4: Thank you again for your valuable suggestion. We have add the figure that describes the block diagram of the closed loop system. The added parts are in page 7, The changes are as follows:

Fig. 2. Advanced Fuzzy II controller for SMB

Question 5: Please add figures that demonstrate the control action for all cases.

Replies 5: Thank you again for your professional suggestion. It has greatly improved our writing skills, and we sincerely appreciate it. We have add the add figures that demonstrate the control action for figure 4, others are similar to it. The added parts are in page 12 marked in red, The changes are as follows:

Taking the control objective illustrated in Figure 4 as an example, the following Figure 6 and 7 shows the progression of force exerted during the control process. It can be observed that the control equipment needs to frequently switch control directions during the process. In practical applications, the requirements for the control equipment are relatively high, as the controller must achieve smooth transitions during these direction changes.

Fig.6. Diagram of Force Application in the Purity Control Process of Material B

Fig.7. Diagram of Force Application in the Purity Control Process of Material A

Question 6: In conclusions, please add an enhancement percentage (%) that demonstrates the proposed algorithm efficiency for your method when used with another method. 

Replies 6: Thank you again for your professional suggestion. In conclusions, we have add an enhancement percentage (%) that demonstrates the proposed algorithm efficiency for your method when used with another method. The add section is on page 15 of the article, marked in red. The add are as follows:

This study primarily investigates the use of advanced fuzzy II controller for regulating the purity of simulated moving bed separations. The research demonstrates that compared to the traditional fuzzy controller and advanced fuzzy I controller, advanced fuzzy II controller achieves higher control precision, albeit at the cost of slower convergence. The advanced fuzzy II controller also maintains stability and high precision when subjected to variations in adsorbent parameters, feed concentration, and switching time. Particularly under switching time variations, advanced fuzzy II controller remains stable, though it exhibits slightly greater fluctuations compared to type I. Compared to the traditional fuzzy controller, the extraction accuracy of Material B improved by an average of 0.35%, while the accuracy of Material A remained relatively stable. In the stability analysis of moving bed parameter variations, the advanced Type II fuzzy controller exhibited greater stability than the Type I controller, with an average precision stability improvement of 0.6%. Overall, advanced fuzzy II controller proves to be superior to both the traditional fuzzy controller and advanced fuzzy I controller.

.

Explanations to Reviewer 1

Thank you very much for the reviewer for the comments and suggestions. We have revised the manuscript according to the comments and now explain to the reviewer as follows:

 Question 1: Please describe in more detail the Summary of this research, thus highlighting its importance to the scientific and academic community.

 Replies 1: Thank you very much for your thorough review. Your feedback has greatly improved our writing skills, and we sincerely appreciate it. We have revised the conclusion part. The revised section is on page 15 of the article, marked in red. The changes are as follows.

 This study primarily investigates the use of advanced fuzzy II controller for regulating the purity of simulated moving bed separations. The research demonstrates that compared to the traditional fuzzy controller and advanced fuzzy I controller, advanced fuzzy II controller achieves higher control precision, albeit at the cost of slower convergence. The advanced fuzzy II controller also maintains stability and high precision when subjected to variations in adsorbent parameters, feed concentration, and switching time. Particularly under switching time variations, advanced fuzzy II controller remains stable, though it exhibits slightly greater fluctuations compared to type I. Compared to the traditional fuzzy controller, the extraction accuracy of Material B improved by an average of 0.35%, while the accuracy of Material A remained relatively stable. In the stability analysis of moving bed parameter variations, the advanced Type II fuzzy controller exhibited greater stability than the Type I controller, with an average precision stability improvement of 0.6%. Overall, advanced fuzzy II controller proves to be superior to both the traditional fuzzy controller and advanced fuzzy I controller. 

Theoretically, the advanced fuzzy II controller offers a novel approach for controlling highly sensitive nonlinear systems, potentially improving control precision by appropriately increasing the fuzzy control quantities in the dynamics. In practical applications, higher precision control can reduce the frequency of experiments and adjustments, thereby lowering production costs. Additionally, it ensures product consistency and high quality. By precisely controlling various parameters in the separation process, the controller can better manage product quality metrics and meet market demands.

Question 2: Please explain in more detail the figures presented in the article, it highlights their importance for the present work.

Replies 2: Thank you again for your professional suggestion. We have add the detail describtion for figures 4 and 5, to improve the readability of the control process, Figures 6 and 7 have been added to show the variation in the forces applied during the control process, along with relevant explanations. The added parts are in page 10-12 which were marked in red. The changes are as follows:

From the outlet of the extracted liquid in Figure 4 and Table 7, in the first set of experiments, the purity targets for controlling material B and material A were set at 94% and 96%, respectively. For material B, the final control results showed that the advanced fuzzy type II controller achieved 94%, the type I controller achieved 94.78%, and the traditional controller achieved 93.13%. The precision of the type II controller was 100%, the type I controller was 99.2%, and the traditional controller was 99.8%. For material A, the type II controller achieved 95.94%, the type I controller achieved 96.14%, and the traditional controller achieved 96.02%. The precision of the type II controller was 99.9%, the type I controller was 99.8%, and the traditional controller was 99.9%.

In the experiments shown in Figure 5 and Table 7, in the second set of experiments, the purity targets for controlling material B and material A were set at 96% and 94%, respectively. For material B, the final control results showed that the advanced fuzzy type II controller achieved 96%, the type I controller achieved 96.5%, and the traditional controller achieved 95.43%. The precision of the type II controller was 100%, the type I controller was 99.4%, and the traditional controller was 99.5%. For material A, the type II controller achieved 94.04%, the type I controller achieved 94.11%, and the traditional controller achieved 94.01%. The precision of the type II controller was 99.9%, the type I controller was 99.8%, and the traditional controller was 99.9%.

Taking the control objective illustrated in Figure 4 as an example, the following Figure 6 and 7 shows the progression of force exerted during the control process. It can be observed that the control equipment needs to frequently switch control directions during the process. In practical applications, the requirements for the control equipment are relatively high, as the controller must achieve smooth transitions during these direction changes.

Fig.6. Diagram of Force Application in the Purity Control Process of Material B

Fig.7. Diagram of Force Application in the Purity Control Process of Material A

Question 3: I suggest inserting one more topic, right after the Conclusion, addressing what is expected from this work in the future. This aspect is essential as it demonstrates the importance of the pre

---

## [Decision Letter · Decision Letter 1]

13 Nov 2024

Purity Control of Simulated Moving Bed Based on Advanced Fuzzy II Controller

PONE-D-24-28958R1

Dear Dr. Chaofan Xie,

We’re pleased to inform you that your manuscript has been judged scientifically suitable for publication and will be formally accepted for publication once it meets all outstanding technical requirements.

Kind regards,

Mazyar Ghadiri Nejad, Ph.D.

Academic Editor

PLOS ONE

Reviewers' comments:

Reviewer's Responses to Questions

**Comments to the Author**

1. If the authors have adequately addressed your comments raised in a previous round of review and you feel that this manuscript is now acceptable for publication, you may indicate that here to bypass the “Comments to the Author” section, enter your conflict of interest statement in the “Confidential to Editor” section, and submit your "Accept" recommendation.

Reviewer #1: All comments have been addressed

Reviewer #2: All comments have been addressed

2. Is the manuscript technically sound, and do the data support the conclusions?

Reviewer #1: Yes

Reviewer #2: Yes

3. Has the statistical analysis been performed appropriately and rigorously? 

Reviewer #1: Yes

Reviewer #2: Yes

4. Have the authors made all data underlying the findings in their manuscript fully available?

Reviewer #1: Yes

Reviewer #2: Yes

5. Is the manuscript presented in an intelligible fashion and written in standard English?

Reviewer #1: Yes

Reviewer #2: Yes

6. Review Comments to the Author

Reviewer #1: Congratulations Dear Authors, after a long analysis it was observed that all corrections were successfully carried out. Therefore, I approve of the current work.

Grateful.

Reviewer #2: The contribution of this revised manuscript is clear and good. All comments from the reviewers were very good, and the author’s answer was very good and satisfied. Therefore, my recommendation is to accept the revised manuscript that has Ref. No.: PONE-D-24-28958R1 for publication.

7. PLOS authors have the option to
publish the peer review history of their article (what does this mean?). If published, this will include your full peer review and any attached files.

Reviewer #1: **Yes: **Gabriel Gomes de Oliveira

Reviewer #2: No

---

## [Editor Report · Acceptance letter]

14 Nov 2024

PONE-D-24-28958R1 

PLOS ONE

Dear Dr. Xie, 

I'm pleased to inform you that your manuscript has been deemed suitable for publication in PLOS ONE. Congratulations! Your manuscript is now being handed over to our production team.

Kind regards, 

on behalf of

Assoc. Prof. Dr. Mazyar Ghadiri Nejad 

Academic Editor

PLOS ONE